# Plasma Lysophosphatidic Acid Concentrations in Sex Differences and Psychiatric Comorbidity in Patients with Cocaine Use Disorder

**DOI:** 10.3390/ijms242115586

**Published:** 2023-10-25

**Authors:** Nerea Requena-Ocaña, María Flores-López, Nuria García-Marchena, Francisco J. Pavón-Morón, Carmen Pedraza, Agustín Wallace, Estela Castilla-Ortega, Fernando Rodríguez de Fonseca, Antonia Serrano, Pedro Araos

**Affiliations:** 1Instituto de Investigación Biomédica de Málaga y Plataforma en Nanomedicina (IBIMA-Plataforma BIONAND), 29590 Málaga, Spain; nerea.requena@ibima.eu (N.R.-O.); maria.flores@ibima.eu (M.F.-L.); javier.pavon@ibima.eu (F.J.P.-M.); mdpedraza@uma.es (C.P.); pedro.araos@ibima.eu (P.A.); 2Unidad de Gestión Clínica de Salud Mental, Hospital Regional Universitario de Málaga, 29010 Málaga, Spain; ngmarchena@ucm.es; 3Departamento de Psicobiología y Metodología en Ciencias del Comportamiento, Facultad de Psicología, Universidad Complutense de Madrid, 28223 Madrid, Spain; 4Unidad de Gestión Clínica del Corazón, Hospital Universitario Virgen de la Victoria de Málaga, 29010 Málaga, Spain; 5Centro de Investigación Biomédica en Red de Enfermedades Cardiovasculares (CIBERCV), Instituto de Salud Carlos III, 28029 Madrid, Spain; 6Departamento de Psicobiología y Metodología de las Ciencias del Comportamiento, Facultad de Psicología, Universidad de Málaga, 29010 Málaga, Spain; awallace@uma.es (A.W.); ecastilla@uma.es (E.C.-O.)

**Keywords:** sex differences, cocaine use disorder, lysophosphatidic acid, psychiatric comorbidity, anxiety disorder, attention-deficit/hyperactivity disorder, personality disorder

## Abstract

We have recently reported sex differences in the plasma concentrations of lysophosphatidic acid (LPA) and alterations in LPA species in patients with alcohol and cocaine use disorders. Preclinical evidence suggests a main role of lysophosphatidic acid (LPA) signaling in anxiogenic responses and drug addiction. To further explore the potential role of the LPA signaling system in sex differences and psychiatric comorbidity in cocaine use disorder (CUD), we conducted a cross-sectional study with 88 patients diagnosed with CUD in outpatient treatment and 60 healthy controls. Plasma concentrations of total LPA and LPA species (16:0, 18:0, 18:1, 18:2 and 20:4) were quantified and correlated with cortisol and tryptophan metabolites [tryptophan (TRP), serotonin (5-HT), kynurenine (KYN), quinolinic acid (QUIN) and kynurenic acid (KYNA)]. We found sexual dimorphism for the total LPA and most LPA species in the control and CUD groups. The total LPA and LPA species were not altered in CUD patients compared to the controls. There was a significant correlation between 18:2 LPA and age at CUD diagnosis (years) in the total sample, but total LPA, 16:0 LPA and 18:2 LPA correlated with age at onset of CUD in male patients. Women with CUD had more comorbid anxiety and eating disorders, whereas men had more cannabis use disorders. Total LPA, 18:0 LPA and 20:4 LPA were significantly decreased in CUD patients with anxiety disorders. Both 20:4 LPA and total LPA were significantly higher in women without anxiety disorders compared to men with and without anxiety disorders. Total LPA and 16:0 LPA were significantly decreased in CUD patients with childhood ADHD. Both 18:1 LPA and 20:4 LPA were significantly augmented in CUD patients with personality disorders. KYNA significantly correlated with total LPA, 16:0 LPA and 18:2 LPA species, while TRP correlated with the 18:1 LPA species. Our results demonstrate that LPA signaling is affected by sex and psychiatric comorbidity in CUD patients, playing an essential role in mediating their anxiety symptoms.

## 1. Introduction

Cocaine use disorder (CUD) is considered one of the main global health problems that entails strong health, social and economic loses in Western societies [1,2]. In Spain, 10.9% of the population between the ages of 15 and 64 have used powdered cocaine at some time, 2.5% in the last year and 1.1% in the last 30 days. Furthermore, cocaine remains the illegal drug that causes the highest number of treatment admissions in Spain (45.4%) [3]. Among the medical consequences associated with long-term cocaine use, we can highlight the prevalence of cardiovascular diseases (e.g., cerebral hemorrhages and myocardial), pulmonary lesions (smoked cocaine) and obstetric problems during pregnancy [4]. Moreover, cocaine is a psychoactive substance that has a direct impact on the Central Nervous System (CNS), leading to neural and psychiatric alterations [4,5,6].

Patients with CUD usually present a high prevalence of psychiatric comorbidity throughout their lives (around 60%), including Axis I or Axis II psychiatric disorders or other substance use disorders [7,8,9]. Mood, psychotic and anxiety disorders are the most prevalent among Axis I disorders, whereas both antisocial and borderline personality disorders are common among Axis II disorders [5]. Among depressive and psychotic disorders, those induced by cocaine are the most prevalent, while anxiety disorders are the most frequent among primary disorders (not induced by cocaine) [5,6]. In addition, CUD patients usually have a high prevalence of alcohol and cannabis use disorders [10]. It is important to note that psychiatric comorbidity in CUD patients is related to a poor prognosis of the disease and less adherence to treatment [5,11]. Furthermore, sex differences between men and women with cocaine abuse has been widely reported [12,13]. Women seem to be more vulnerable to the reinforcing effects of cocaine as well as the development of depression, anxiety, coping with stress, and activation of the hypothalamic–pituitary–adrenal (HPA) axis. The effects can be partially explained by biological differences in male and female gonadal hormones and in chromosomal mechanisms (X and Y) [14]. Thus, despite women having lower rates of cocaine use and attending treatment centers less than men, they suffer more anxiety, mood and eating disorders [13,15]. However, antisocial personality disorders and other concomitant substance use disorders—including tobacco, alcohol and cannabis—are more common among men with CUD [6,16]. In addition, women suffer more social discrimination and worse educational levels, unemployment rates, socioeconomic status and health conditions. Instead, men have a higher prevalence of legal or criminal problems [13,17,18]. Therefore, given that different therapeutic strategies are necessary in men and women due to their different biological and social conditions, it is necessary to focus on sex differences in substance use disorders to create new effective treatments [12,18,19].

In an attempt to further explore potential biological molecules that could help determine the effects of cocaine on sex differences and psychiatric comorbidity, we selected lysophosphatidic acid (LPA). LPA is a small endogenous bioactive glycerophospholipid that is present in blood plasma at higher concentrations. The autotaxin enzyme is the main precursor for the synthesis of LPA and catalyzes the elimination of the choline group of lysophosphatidylcholine (LPC) to form LPA [20,21]. LPA exerts its actions through the G-protein-coupled receptors LPA_1-6_ that are expressed in a wide variety of tissues and regulate several important biological processes (i.e., wound healing, cancer, obesity and fibrosis) [22]. Furthermore, LPA has different endogenous chemical species identified by the length and degree of saturation of the fatty acid moiety that include palmitic (16:0), stearic (18:0), oleic (18:1), linoleic (18:2) and arachidonic acids (20:4). Those LPA species are the most abundant in human plasma and serum [23,24]. Thus, LPA receptors have been shown to be differentially activated by LPA species. LPA_1_ and LPA_2_ receptors show a broad ligand specificity with LPA species with some differences. However, the LPA_3_ receptor has the highest reactivity for LPA species 18:1, 18:2 and 18:3, followed by LPA species 16:1 and 20:4 [25]. LPA signaling has an essential role in the CNS due to its expression in several brain cells (i.e., neurons, astrocytes, microglia and oligodendrocytes) that affect development, survival, migration, myelination and cell proliferation [22,26]. Thus, LPA receptors have been implicated in neurodevelopmental and neuropsychiatric diseases such as neuropathic pain, ischemic stroke, traumatic brain injury, spinal cord injury, fetal hypoxia, fetal hydrocephalus, schizophrenia and Alzheimer disease [22,27]. Furthermore, sex differences in all LPA species unless for LPA 18:0 and 18:1 have been found in previous studies revealing a robust sexual dimorphism [23,28,29]. On the other hand, because the activation of the LPA receptor has been involved with the release of neurotransmitters (e.g., dopamine) associated with behavior, reward and learning and memory processes, it has been suggested to be implicated in substance use disorders [30,31,32,33]. Thus, we have recently found that patients with CUD have significantly lower plasma concentrations of total LPA and some LPA species [29,34], while the mRNA expression of the LPA_1_ receptor has been found to be higher than controls [29]. In addition, several preclinical studies have also linked the effects of LPA with anxiety like-behaviors [33]. Anxiety, memory impairments, hippocampal dysfunction and motor alterations have been reported in studies with LPA_1_ knockout mice [27]. However, little is known about the possible modulation of LPA and its species over psychiatric disorders in humans.

In this context, stress is the main precursor of affective disorder vulnerability through the activation of the HPA axis and the release of glucocorticoids [35]. The adaptative function of the cortisol response consists of a rapid rise and decline following an environmental stressor. A prolonged or excessive cortisol response due to repetitive stress, especially in the early stages of life, may lead to the development of a mental disorder [36]. Paradoxically, low cortisol levels are shown in psychiatric conditions such as major depression, bipolarity, anxiety and schizophrenia [37,38,39]. However, cortisol levels might change in opposite directions depending on the subtype of depressive disorder (melancholic vs. atypical/psychotic) or in the phase of bipolarity (mania/irritability) [37]. In addition, there are sex differences in cortisol changes. Woman with major depressive disorder or anxiety disorder have an attenuated cortisol stress response, whereas men with major depressive disorder and social anxiety disorder exhibit an augmented cortisol response [39]. On the other hand, increased corticosteroids concentrations have been associated with the dysregulation of the kynurenine pathway, by which the amino acid tryptophan (TRP) is metabolized [40,41,42]. TRP is the precursor of serotonin (5-HT), but it is primarily metabolized to kynurenine (KYN) and then in the quinolinic acid (QUIN) and the kynurenic acid (KYNA). Several studies have found that TRP metabolites might play a role in stress and anxious behavior: while KYN and QUIN exert neurotoxic and anxiogenic responses, KYNA has neuroprotective and anxiolytic properties [43,44]. Recently, we have linked CUD with lower plasma concentrations of KYNA; however, serotonin (5-HT) concentrations were increased in CUD patients with depressive and personality disorders [45].

Therefore, the main objectives of the present observational study were to investigate (1) sex differences in cocaine-related variables and psychiatric comorbidity, (2) the impact of cocaine abuse on plasma concentrations of LPA species compared to controls, (3) sex differences in plasma concentrations of LPA species (sexual dimorphism), (4) the differences in plasma concentrations of LPA species among psychiatric disorders in CUD patients, and (5) the associations between plasma levels of LPA species and stress-related factors such as cortisol and TRP metabolites (TRP, KYN, QUIN, KYNA, 5-HT).

## 2. Results

### 2.1. Sociodemographic Characteristics

Table 1 shows a socio-demographic description of the total sample. We recruited 88 patients in abstinence from cocaine use outpatient programs and 60 healthy control subjects. The mean age of the CUD group was 37 years, and 84% of the participants were men with a BMI index of 26 kg/m^2^. Similarly, the mean age of the control group was 38 years, and 83% were men with a BMI of 26 kg/m^2^. However, significant differences were observed between both groups with respect to educational level (*p* < 0.001) and occupation (*p* < 0.001).

### 2.2. Prevalence and Sex Differences in Cocaine-Related Variables, Psychiatric and Medical Comorbidity, and Use of Medication in Abstinent CUD Patients

The variables related with the CUD group were evaluated and described in Table 2. The mean age at first cocaine use was 18 years, the average age of the CUD onset was 24 years and the length of CUD throughout life was 7 years. The mean of the severity criteria of addiction was seven [based on Fifth Edition (DSM-5)], and they had a length of 43 days of abstinence at the moment of the evaluation.

We observed an elevated prevalence of other comorbid psychiatric disorders (59%), showing lifetime mood, anxiety, personality and childhood attention-deficit/hyperactivity (ADHD) disorders as the most frequently diagnosed in a range from 30% to 18%. Regarding other abused substances, there was a high prevalence of other substance use disorders (SUDs; 57%), alcohol and cannabis were the most prevalent substances of abuse (53% and 31%, respectively). In addition, 63% of the abstinent cocaine patients received psychiatric medication during the last year, especially anxiolytics (42%) and antidepressants (35%). Furthermore, 17% of the CUD group had a medical problem.

Gender differences were analyzed in CUD patients using the Mann–Whitney U test for cocaine-related variables. Fisher’s exact test was used for variables related to psychiatric and medical comorbidity and psychiatric medication. As we expected, men began using cocaine earlier than women (*p* < 0.027). Women had a higher prevalence of anxiety and eating disorders (χ^2^ = 12.41, *p* < 0.001; χ^2^ = 16.09, *p* < 0.002), whereas men had more cannabis use disorder (χ^2^ = 7.28, *p* = 0.007). In addition, women used more prescribed anticraving medication than men (χ^2^ = 6.05, *p* = 0.014).

**Table 2 ijms-24-15586-t002:** Psychiatric characteristics of the CUD group and sex differences.

Variables		CUD Group				
		Total N = 88	Men N = 74	Women N = 14	Statistic	*p*-Value
Age at first cocaine use ^1^ [Mean (SD)]	Years	17.53 (7.93)	19.13 (0.69)	22 (1.51)	210	**0.027**
Age at onset of CUD ^1^ [Mean (SD)]	Years	23.65 (10.60)	26.98 (1.04)	27.25 (1.55)	333.500	0.293
Length of CUD diagnosis ^1^ [Mean (SD)]	Years	6.71 (6.74)	7.65 (0.94)	6.92 (1.86)	413	0.708
Severity criteria ^1^ [Mean (range)]	Criteria [1,2,3,4,5,6,7,8,9,10,11]	7.28 (3.20)	7.09 (0.39)	8.29 (0.54)	433	0.328
Length of abstinence ^1^ [Mean (mode)]	Days	42.90 (72.56)	40.35 (8.86)	52.83 (20.34)	340.500	0.149
Comorbid substance use disorders ^2^ [N (%)]	Alcohol	46 (52.30)	39 (52.70)	7 (50)	0.034	0.853
	Cannabis	27 (30.70)	27 (36.50)	-	7.285	**0.007**
	Sedatives	7 (8)	7 (9.50)	-	1.422	0.233
	Opiates	5 (5.70)	5 (6.80)	-	0.992	0.319
	Stimulants	4 (4.50)	3 (4.10)	1 (7.10)	0.259	0.613
Comorbid psychiatric disorders ^2^ N (%)]	Mood	26 (29.50)	23 (31.10)	3 (21.40)	0.521	0.470
	Anxiety	23 (26.10)	14 (18.90)	9 (64.30)	12.408	**<0.001**
	Personality	20 (22.7)	16 (21.60)	4 (28.6)	0.320	0.572
	ADHD	16 (18.20)	15 (20.30)	1 (7.10)	1.348	0.243
	Psychotic	6 (6.80)	6 (8.10)	-	1.204	0.272
	Eating	5 (5.70)	1 (1.40)	4 (28.60)	16.09	**<0.001**
Psychiatric medication ^2^ [N (%)]	Anxiolytics	37 (42)	30 (40.50)	7 (50)	0.732	0.392
	Antidepressants	31 (35.20)	26 (35.10)	5 (35.70)	0.038	0.845
	Disulfiram	14 (15.90)	13 (17.80)	1 (7.70)	0.819	0.365
	Antipsychotics	9 (10.20)	7 (9.50)	2 (14.30)	0.391	0.532
	Anticraving	6 (6.80)	3 (4.10)	3 (21.40)	6.046	**0.014**
Medical problem	No Yes	82.07 17.30	10 (13.50)	4 (28.60)	2.507	0.113

^1^ Value was calculated with Mann–Whitney U test. ^2^ Value was calculated with Fischer’s exact test or chi-squared test. Bold values are statistically significant for *p* < 0.05. Abbreviation: ADHD = attention-deficit/hyperactivity disorder (childhood), CUD = cocaine use disorder.

### 2.3. Plasma Concentrations of LPA Species in Abstinent CUD Patients

The impact of cocaine abuse on plasma concentrations of LPA was analyzed in the total sample using two-way ANCOVA with “group” (CUD group and control group) and “sex” as factors and age and BMI as covariates.

As shown in Figure 1A, we did not find an effect of the “group” factor in total LPA, 16:0 LPA, 18:0 LPA, 18:1 LPA, 18:2 LPA and 20:4 LPA (Appendix A). However, plasma concentrations of total LPA, 16:0 LPA, 18:1 LPA, 18:2 LPA and 20:4 LPA [F(1,142) = 12.84; *p* < 0.001, F(1,142) = 17.76; *p* < 0.001, F(1,142) = 8.39; *p* = 0.004, F(1,142) = 11.59; *p* = 0.001, and F(1,142) = 6.945; *p* = 0.009] were significantly different between men and women (Appendix A). Women had higher concentrations of total LPA, 16:0 LPA, 18:1 LPA, 18:2 LPA and 20:4 LPA than men (Figure 1B). We did not find an interaction effect between “group” and “sex” factors in total LPA and its species.

### 2.4. Correlation Analysis between Plasma Concentrations of LPA and Cocaine-Related Variables

Furthermore, correlation analyses using Spearman partial correlations controlling for sex and BMI were performed between plasma concentrations of total LPA and LPA species with cocaine-related variables (age at first cocaine use, age at onset of CUD, length of CUD, length of abstinence and severity criteria).

We only observed a negative and significant correlation between plasma concentrations of 18:2 LPA and age at onset of CUD (years; rho = −0.284, *p* = 0.032; Appendix A). However, when men and women were analyzed separately, we found that age at onset of CUD was negative and significant correlated with total LPA (rho = −0.252, *p* = 0.050), 16:0 LPA (rho = −0.279, *p* = 0.029) and 18:2 LPA (rho = −0.283, *p* = 0.027), only in male patients (Figure 2). We did not find significant correlations between LPA and cocaine-related variables in women.

### 2.5. Plasma Concentrations of LPA Species in Comorbid Psychiatric Disorders in Abstinent CUD Patients

As shown in the clinical description of the sample in Table 2; mood, anxiety, personality and childhood ADHD disorders were the most frequent comorbid psychiatric disorders in the CUD group. Thus, we investigated the effect of these disorders in the total LPA and LPA species in the CUD group using a two-way ANCOVA with “psychiatric comorbidity” (e.g., “comorbid mood disorder vs. no comorbid mood disorder”) and “sex” as factors and age and BMI as covariates.

As shown in Figure 3A, we did not observe a main effect of “comorbid mood disorder” on LPA concentrations (Appendix A). Nevertheless, plasma concentrations of total LPA, 16:0 LPA, 18:2 LPA and 20:4 LPA [F(1,82) = 6.45; *p* = 0.013, F(1,82) = 7.54; *p* = 0.007, F(1,82) = 7.25; *p* = 0.009, F(1,82) = 4.22; *p* = 0.043] were significantly affected by the “sex” factor, with women having higher concentrations than men. However, we did not find an interaction effect on LPA concentrations.

Plasma concentrations of total LPA, 18:0 LPA and 20:4 LPA were affected by “comorbid anxiety disorder” (Figure 3B). Thus, CUD patients with comorbid anxiety disorder had lower concentrations of total LPA, 18:0 LPA and 20:4 LPA species [F(1,82) = 6.03; *p* = 0.016, F(1,82) = 5.75; *p* = 0.019, and F(1,82) = 4.58; *p* = 0.035] compared to those without an anxiety disorder (Appendix A). Moreover, plasma concentrations of total LPA, 16:0 LPA, 18:1 LPA, 18:2 LPA and 20:4 LPA [F(1,82) = 17.84; *p* < 0.001, F(1,82) = 15.97; *p* < 0.001, F(1,82) = 7.22; *p* = 0.009, F(1,82) = 12.30; *p* = 0.001, and F(1,82) = 13.06; *p* = 0.001] were significantly affected by the “sex” factor, with women having higher concentrations than men. We also observed an interaction effect between “comorbid anxiety disorder” and “sex” on 20:4 LPA [F(1,82) = 4.75; *p* = 0.032], and this was also marginally significant for total LPA [F(1,88) = 3.92; *p* = 0.051]. As shown in Figure 4, women without anxiety had higher concentrations of total LPA and 20:4 LPA than men with and without anxiety (*p* = 0.005, *p* = 0.001; *p* = 0.001, *p* < 0.001, respectively).

Plasma concentrations of 18:1 LPA and 20:4 LPA species were affected by “comorbid personality disorder” (Figure 3C). Thus, CUD patients with comorbid personality disorders had higher concentrations of 18:1 LPA and 20:4 LPA species [F(1,82) = 5.31; *p* = 0.024, and F(1,82) = 8.54; *p* = 0.004] compared with those without comorbid personality disorders (Appendix A). Moreover, plasma concentrations of total LPA, 16:0 LPA, 18:1 LPA, 18:2 LPA and 20:4 LPA [F(1,82) = 9.21; *p* = 0.003, F(1,82) = 12.02; *p* = 0.001, F(1,82) = 5.92; *p* = 0.017, F(1,82) = 4.42; *p* = 0.039, and F(1,82) = 8.54; *p* = 0.004] were significantly affected by the “sex” factor, with women having higher concentrations than men. However, we did not find an interaction effect on LPA concentrations.

Plasma concentrations of total LPA and 20:4 LPA species were affected by “comorbid ADHD disorder” (Figure 3D). Thus, CUD patients with comorbid ADHD disorder had significantly lower concentrations of total LPA and 16:0 LPA [F(1,82) = 4.93; *p* = 0.029, and F(1,82) = 6.07; *p* = 0.016] compared with those without ADHD (Appendix A). However, we did not find sex or interaction effects on LPA concentrations.

### 2.6. Plasma Concentrations of LPA Species in Comorbid Alcohol and Cannabis Use Disorders in Abstinent CUD Patients

As alcohol and cannabis use disorders were highly prevalent in CUD patients (show Table 2), we wanted to investigate if their comorbidity could affect to concentrations of LPA species in the cocaine abstinent patients using two-way ANCOVA with “comorbid substance use disorder” (e.g., comorbid alcohol use disorder and no comorbid alcohol use disorder) and “sex” as factors and age and BMI as covariates. Nevertheless, despite showing the effect of sex on species of LPA (except for LPA 18:0), we did not observe a main effect of “comorbid alcohol/cannabis use disorder” nor an interaction between both factors on LPA concentrations (Appendix A).

### 2.7. Plasma Concentrations of LPA Species in Comorbid Medical Problems and Psychiatric Medication in Abstinent CUD Patients

Furthermore, we wanted to explore if comorbid medical problems or using psychiatric medication in the last year could affect to concentrations of LPA species in the cocaine abstinent patients using two-way ANCOVA with “comorbid medical problem/psychiatric medication” (comorbid and non-comorbid subgroup/psychiatric medication and no psychiatric medication) and “sex” as factors and age and BMI as covariates. Nevertheless, despite showing the effect of sex on species of LPA (except for LPA 18:0), we did not observe a main effect of “comorbid medical problem/psychiatric medication” nor an interaction between both factors on LPA concentrations (Appendix A, respectively).

Moreover, we wanted to investigate if using anxiolytics in the last year specifically modified plasma concentrations of LPA with “anxiolytic medication/ no anxiolytic medication” and “sex” as factors and age and BMI as covariates. However, despite observing the effect of sex on species of LPA (except for LPA 18:0), we did not find a significant effect of “anxiolytic medication” nor an interaction between both factors on LPA concentrations (Appendix A).

### 2.8. Correlations Analysis between Plasma Concentrations of LPA, Cortisol and Tryptophan Metabolites in Abstinent CUD Patients

Moreover, in order to explore the potential anxiolytic effect of LPA, we performed correlation analyses using Spearman partial correlations controlling for sex, age and BMI between plasma concentrations of total LPA and LPA species with cortisol and tryptophan metabolites (TRP, KYN, KYNA, QUIN, 5-HT).

As shown in Figure 5, we found robust positive and significant correlations between total LPA, 16:0 LPA and 18:2 LPA species with KYNA (rho = 0.668, *p* = 0.002, rho = 0.713, *p* < 0.001, rho = 0.606, *p* = 0.006, respectively). There was also a positive and significant correlation between 18:1 LPA and TRP (rho = 0.519, *p* = 0.023). However, we did not observe significant correlations between LPA species and cortisol, KYN, QUIN and 5-HT (Appendix A).

## 3. Discussion

While the association between LPA signaling and an anxiety phenotype has been well established in preclinical studies, the contribution of this biomolecule in psychiatric conditions in humans is poorly investigated. We lack relevant information on how LPA signaling influences neuroadaptations resulting from a lifetime of cocaine use, the high prevalence of psychiatric comorbidity and sex differences.

Preclinical studies of cocaine conditioning have demonstrated specific changes in the lipidomic profile of rat blood and brain—especially in the hippocampus—that were associated with the initial response to cocaine or locomotor sensitization [46,47]. Recently, studies in rats showed that plasma levels of total LPA were lower after 30 min of acute and chronic cocaine administration (5 mg/kg for males and 30 mg/kg for females). In addition, total LPA concentrations were lower before 72 h of abstinence in male rats that were exposed to 15 mg/kg every day for two weeks, but abstinence had no impact on total LPA levels in females rats [29]. In contrast, intracerebroventricular infusions of LPA in mice might increase adult-hippocampal neurogenesis that enhances forgetting of cocaine-associated contextual memories [48]. On the other hand, lower plasma levels of total LPA, LPA 18:2 and LPA 16:0 have been reported in patients with substance use disorders [34]. Especially, decreases in total LPA, 16:0 LPA, 18:1 LPA, 18:2 and 20:4 have been found in CUD patients compared to controls [29]. Furthermore, plasma concentrations of total LPA, 18:1, 18:2 and 20:4 were even more decreased in CUD patients compared to patients with alcohol use disorder [34]. However, despite this study did not finding altered plasma concentrations of LPA in patients with alcohol use disorder, other studies have described decreased plasma LPA levels compared with controls [28,49]. Plasma LPA concentrations could even be a reliable biomarker of alcoholic liver disease (e.g., fatty liver/steatosis, alcoholic steatohepatitis or cirrhosis) [49]. Nevertheless, in this study we did not find statistical differences in plasma LPA concentrations in CUD patients. Although the nature of this result is unknown, one possible explanation might be that we included patients with cocaine as the main drug for which treatment was sought, and not only by CUD diagnosis. Another possibility could be the small sample size compared to those studies. However, we found that 18:2 LPA was negatively correlated with the age of CUD diagnosis in the total sample, with total LPA, 16:0 LPA and 18:2 LPA being correlated with the age of CUD onset, especially in men. This result may indicate a compensatory response against the early development of problematic cocaine use since LPA species have been considered as neuroprotective for signaling that correlates with Brain-Derived Neurotrophic Factor (BDNF) in abstinent alcohol patients [28]. Therefore, we need further investigation to unravel the role of LPA in substance use disorders, especially for cocaine and alcohol use disorders and their comorbidity.

As we expected, we observed that plasma concentrations of total LPA and some species of LPA were affected by sex in both the control group and the CUD group. Thus, women exhibit higher plasma concentrations of total LPA and LPA species—including 16:0 LPA, 18:1 LPA, 18:2 LPA and 20:4 LPA—compared with men, which suggests that sex has a high impact on the circulating species of this lipid. These results are consistent with previous studies in healthy subjects that have reported sex differences with higher LPA concentrations in women [50,51]. The autotaxin–LPA pathway could also develop an essential role in endocrine function since several studies have proposed them as potential biomarkers of hepatic steatosis in obesity [52] as well as in breast, ovarian and endometrial cancer in women [53,54,55]. Therefore, LPA concentrations could be linked to female hormones or differences in the Y chromosome that could explain their psychiatric vulnerability especially in anxiety disorders.

On the other hand, CUD patients usually display a high prevalence of psychiatric comorbidities associated with other substance use disorders and other mental disorders. Our results indicate that mood, anxiety and personality disorders as well as alcohol and cannabis use disorders were the most frequent among abstinent CUD patients, as many other studies have reported [9,13,56,57,58,59]. Regarding its association with sex-based differences, in our study, women began using cocaine later and exhibited higher rates of anxiety and eating disorders, whereas men showed more cannabis use disorders. In the same line, previous studies have suggested that women suffer more with generalized anxiety disorder, post-traumatic stress disorder, anorexia and bulimia, which may be considered risk factors for the development of CUD. In contrast, men are more likely to develop other substance use disorders—except for sedatives [6,16,59]. Thus, alcohol use has been biologically demonstrated to increase the risk of compulsive cocaine consumption [60]. However, although we found a high rate of cannabis use disorder in men compared to women, we did not observe that men had more alcohol use disorders as many other studies have reported. In this context, because LPA has been widely linked with stress and anxiety responses in mice [33], we also evaluated the existence of possible associations between circulating LPA levels and the presence of comorbid psychiatric or substance use disorders throughout life in CUD patients.

The autotaxin–LPA pathway modulates cortical glutamatergic hyperexcitability and its disruption has been associated with neuropsychiatric diseases such as altered analgesia, schizophrenia, Alzheimer disease and multiple sclerosis [61]. There is a high LPA_1_ receptor density in emotion-related regions such as the frontal cortex, hippocampus, amygdala and striatum [62,63]. Thus, the LPA–LPA_1_ pathway develops an essential role in emotion and stress coping behaviors, which is strongly associated with the psychopathological endophenotype of depression and anxiety [64]. Upon exposure to acute stress, a perturbation in the concentrations of both total LPA and LPA species in the hippocampus of mice is observed [65]. Mice lacking the LPA_1_ receptor display augmented activation of the limbic system, exhibiting increased anhedonic behavior, hypoactivity, stress reactivity and anxiety responses [33,66]. In addition, this genotype shows reduced fear extinction, motivation and associative spatial memory compared with wild type mice [55,59,60,61]. LPA_1_-null mice also show an altered adult neurogenesis and cell apoptosis in the hippocampus and cortical regions [33,63,66]. In agreement, we found that total LPA and some of its species are downregulated in CUD patients with comorbid anxiety disorders (18:0 LPA and 20:4 LPA) and ADHD (16:0 LPA). It was expected that LPA concentrations vary in the same direction, since anxiety disorders and ADHD frequently overlap in both children and adults (e.g., generalized or social) [67,68,69,70]. Interestingly, we observed an interaction effect between anxiety and sex in total LPA and 20:4 LPA. This result suggests that women may have an anxiety disorder when their circulating levels of LPA are similar to those in men. Therefore, since women with anxiety disorders have lower levels of plasma LPA concentrations, pharmacological therapies aimed at increasing its levels could be a promising treatment to alleviate symptoms of stress and anxiety. In this regard, because preclinical studies have demonstrated that LPA_1_ knockout mice consume alcohol to mitigate their anxiety [71], men and women with low circulant levels of LPA may be particularly vulnerable to alcohol, tobacco, cannabis and hypno-sedative addiction. In addition, dimorphism in plasma LPA concentrations (men have lower levels) could also explain why men have higher rates of alcohol and cannabis use disorders than women.

On the other hand, we found higher plasma concentrations of 18:1 LPA and 20:4 LPA in CUD patients with personality disorder (cluster B)—composed by antisocial (45.84%) plus borderline (54.16%). Therefore, our results suggest that while lower plasma LPA levels are associated with anxiety and ADHD, higher LPA levels are associated with the cluster B of personality disorders (dramatic/emotional/erratic) that are not specially associated with anxiety traits [72]. In contrast, cluster C personality disorders (anxious–fearful) are the most frequently related to anxiety comorbidity, occurring more than twice as often as cluster A or B personality disorders [73]. Interestingly, preclinical and clinical studies have described higher concentrations of 18:1 and 20:4 LPA species in spinal cord and cerebrospinal fluid in neuropathic pain [74,75], which is associated with cluster C personality disorders [76,77]. Nevertheless, more investigation about the role of LPA on personality disorders is needed.

It is important to highlight that despite some preclinical and clinical studies having reported associations between a depressive-anxiety phenotype and LPA [66,78], we did not observe significant alterations in CUD patients with mood disorders as other researchers have described [79,80]. However, this might be explained because our CUD sample was diagnosed separately for mood and anxiety disorders, which means that our depressive group probably lacks severe anxiety symptoms.

Finally, in the present study, we also evaluated the correlation between LPA species and biological factors related to stress and anxiety, including cortisol and TRP metabolites. Thus, we observed that higher circulating concentrations of TRP were associated with higher plasma levels of LPA 18:1 species, which in turn were observed in CUD patients with personality disorders. Elements of the pathology of personality disorders may be due in part to serotonergic dysfunction. In this regard, antisocial personality patients had reported augmented TRP plasma levels observing a disturbed TRP metabolism associated with violent behavior [72,73,81]. Similarly, a tryptophan-hydroxylase 1 and 2 (TPH 1 and 2) “risk” haplotype has been described to be associated with borderline personality, with affective lability, suicidal behavior and aggression symptomatology [82,83]. On the other hand, the KYN pathway is involved in the catabolism of tryptophan into other subproducts in the brain through microglia and astrocytes such as KYNA and QUIN. These biomolecules can alter excitatory neurotransmission and mediate the neuroimmune system. Thus, KYNA has been related to neuroprotection because of its ability to interrupt glutamatergic neurotransmission and their antioxidant and anti-inflammatory properties [44,84]. Ketamine treatment, a potent sedative and antidepressant, has been associated with increases in KYNA levels in patients with bipolar disorder and schizophrenia [85,86]. Concordantly, decreased plasma concentrations of KYNA and the KYNA/QUIN rate have been related to more depressive and anxiety symptoms [87,88]. Furthermore, reduced serum concentrations of KYNA have also been observed in adults with ADHD [89] and in patients with CUD [45]. Therefore, since we have found strong and positive correlations between total LPA and 16:00 and 18:2 LPA species with KYNA, the anxiety effects derived from low concentrations of LPA are further corroborated.

### Limitations and Future Directions

We are aware of the limitations of the present observational study and the necessity for further investigation. First, larger samples of male and female CUD patients should be included in the comorbid psychiatric groups to disentangle the sex differences in each condition. Second, we did not control variables such as the nutritional diet or the physical exercise of the participants that could influence plasma LPA concentrations. Third, the recruitment of the sample was conducted from outpatient programs, and we did not consider other factors such as nutritional and medical problems that could affect the validity of the results. Fourth, longitudinal studies are essential to monitor changes in LPA and tryptophan metabolites during abstinence over time.

## 4. Materials and Methods

### 4.1. Participants and Recruitment

The present study included 148 Caucasian volunteers divided into two groups: 88 abstinent cocaine use disorder patients in outpatient treatments and 60 control subjects matched by age, body mass index (BMI) and proportion of sex with the cocaine group. Patients were recruited at Centro provincial de Drogodependencias (Málaga, Spain). Control participants were included from databases of healthy subjects willing to participate in medical research projects from Hospital Regional Universitario de Málaga (Málaga, Spain).

To be eligible for the present study, participants had to meet the following inclusion criteria: ≥18 years to 65 years of age and abstinence from cocaine for at least 4 weeks. The exclusion criteria included personal history of long-term inflammatory diseases or cancer, cognitive or language limitations, pregnant or breast-feeding women, and infectious diseases. With regard to the control group, participants with psychiatric disorders in Axis I were also excluded.

### 4.2. Ethics Statements

Written informed consent was obtained from each participant after a complete description of the study. All the participants had the opportunity to discuss any questions or issues. The study and protocols for recruitment were approved by the Ethics Committee of the Hospital Regional Universitario de Málaga (1857-N-20) in accordance with the Ethical Principles for Medical Research Involving Human Subjects adopted in the Declaration of Helsinki by the World Medical Association (64th WMA General Assembly, Fortaleza, Brazil, October 2013), Recommendation No. R (97) 5 of the Committee of Ministers to Member States on the Protection of Medical Data (1997), Spanish data protection act [Regulation (EU) 2016/679 of the European Parliament and of the Council 27 April 2016 on the protection of natural persons with regard to the processing of personal data and on the free movement of such data, and repealing Directive 95/46/EC (General Data Protection Regulation). All collected data were given code numbers in order to maintain privacy and confidentiality.

### 4.3. Clinical Assessments

Substance use disorders and other psychiatric disorders were diagnosed according to the Diagnostic and Statistical Manual of Mental Disorders (DSM-IV-TR) criteria using the Spanish version of the Psychiatric Research Interview for Substance and Mental Disorders (PRISM). PRISM is a semi-structured interview with good psychometric properties in the evaluation of substance use disorders and in the main psychiatric comorbid disorders related to substance use population [90,91].

### 4.4. Collection of Plasma Samples

Blood samples were obtained in the morning after fasting for 8–12 h (prior to the psychiatric interviews). Venous blood was extracted into 10 mL K2 EDTA tubes (BD, Franklin Lakes, NJ, USA) and immediately processed to obtain plasma. Blood samples were centrifuged at 2200× *g* for 15 min (4 °C) and individually assayed to detect infectious diseases using 3 commercial rapid tests for HIV, hepatitis B, and hepatitis C (Strasbourg, Cedex, France). Finally, plasma samples were individually characterized, registered, and stored at −80 °C until further analyses.

### 4.5. Determination of LPA Species

The LPA species of saturated fatty acids palmitic acid (16:0-LPA) and stearic acid (18:0-LPA), the LPA of the monounsaturated fatty acid oleic acid (18:1-LPA) and the LPA species of the polyunsaturated fatty acids linoleic acid (18:2-LPA) were determined using an extraction protocol followed by LC-MS/MS separation and quantification. Quantification of LPA species in human plasma was performed using an ACQUITY UPLC system (Waters Associates, Milford, MA, USA) for the chromatographic separation coupled to a triple quadrupole (Xevo TQ-S micro) mass spectrometer provided with an orthogonal Z-spray-electrospray interface (ESI) (Waters Associates, Milford, MA, USA), as described previously [28]. Plasma concentrations of LPA species were expressed as nanograms of protein per milliliter of plasma (ng/mL) and LPA total was calculated by adding the concentrations of the measured LPA species.

### 4.6. Determination of TRP, KYN, KYNA, QUIN and 5-HT

We wanted to investigate the potential anxiolytic role of LPA throughout its correlation with tryptophan metabolites in 21 cocaine abstinent patients, as previously described [45]. Plasma aliquots (100 μL) were deproteinized by addition of 10 μL of 35% perchloric acid, and tyrosine (15 μL of 50 μM solution) was added as internal standard. The acidified plasma was vortexed, kept at room temperature for 10 min and then centrifuged at 16,000× *g* for 15 min at 4 °C. For KYN measurements, 20 μL of the supernatant was injected onto a reverse phase column (HR-80; 80 mm × 4.6 mm, 3 μm; Thermo Fisher Scientific. Waltham, MA, USA) and KYN was isocratically eluted using a mobile phase containing 0.1 M sodium acetate and 4% acetonitrile (adjusted to pH 4.6) at a flow rate of 1 mL/min (Waters 515 HPLC Pump). KYN was determined via UV detection (360 nm, Waters 2487). TRP and KYNA were separated using the same column as for the determination of KYN but using a mobile phase containing 0.5 M sodium acetate (adjusted to pH to 6.2), 0.25 M zinc acetate and 5% acetonitrile, delivered at 1 mL/min and detected fluorometrically at excitation/emission wavelengths of 270/360 nm for TRP and 344/398 nm for KYNA (Waters 2475, Multi Fluorescence Detector). An ODS2-C18 column (150 mm × 4.6 mm, 5 μm, Waters Spherisorb) was used for the separation of 5-HT. The mobile phase consisted of 0.1 M sodium acetate (adjusted to pH 3.8) and 8% methanol, and 5-HT was detected fluorometrically at excitation/emission wavelengths of 290/337 nm. QA was determined using a commercially available ELISA immunoassay (Cloud-Clone Corp., Houston, TX, USA) according to instructions of the manufacturer.

### 4.7. Determination of Cortisol

We wanted to explore the potential anxiogenic role of LPA throughout its correlation with cortisol in 34 cocaine abstinent patients. Plasma cortisol concentrations were detected using the highly sensitive enzyme-linked immunosorbent assay (ELISA) Kit for cortisol (Cor) (Cat No.: HEA462Ge, Cloud-Clone Corp). Aliquots of each plasma sample (1:2 dilution) were used for the assessment of cortisol based on the instructions of the producer (Cloud Clone, Houston, TX, USA). Specifically, we added 50 μL of the standard or 1:2 dilution sample to each well and then we added 50 μL of prepared detection reagent A immediately. Then, we incubated the plate 1 h at 37 °C. After washing the plate three times, we added 100 μL of prepared detection reagent B. We incubated the plate for 30 min at 37 °C, and then, we washed the plate 5 times after. Subsequently, we added 90 μL of substrate solution and the plate was incubated for 10–20 min at 37 °C again. Finally, we added 50 μL of stop solution and the plate was read at 450 nm immediately.

### 4.8. Statistical Analysis

All data in the tables are expressed as number and percentage of subjects [N (%)] or mean and standard deviation (SD). The significance of differences in categorical and non-normal continuous variables was determined using Fisher’s exact test (chi-square test, χ^2^) and Mann–Whitney U’s test, respectively. Multiple analysis of covariance (ANCOVA) was performed to indicate the relative effects of the explanatory variable (i.e., history of cocaine use disorders and mood disorders) on plasma concentrations of LPA species, controlling for additional independent variables and covariates [sex, age and BMI]. The post hoc tests for multiple comparisons were performed using Sidak’s correction test. Logarithm (10) transformation for dependent variables was used to ensure statistical assumptions for positively skewed distributions (i.e., log concentrations of LPA species). Estimated marginal means and 95% confidence intervals [(95%)] of LPA species were expressed and represented in the figures after back transformations. Correlation analyses were performed using Spearman partial correlations adjusted by covariates [age and BMI]. The statistical analyses were carried out using GraphPad Prism version 5.04 (GraphPad Software, San Diego, CA, USA) and IBM SPSS Statistical version 22 (IBM, Armonk, NY, USA). A *p*-value <0.05 was considered statistically significant.

## 5. Conclusions

We concluded that plasma concentrations of LPA species are related to sex differences and psychiatric comorbidity in abstinent CUD patients treated in an outpatient setting. The main findings are as follows: (i) women exhibit a higher prevalence of anxiety disorders and eating disorders, while men began cocaine use earlier and displayed higher rates of cannabis use disorders; (ii) plasma concentrations of LPA species were not affected by chronic cocaine consumption through life when CUD patients were compared to control subjects, but some LPA species (total LPA, 16:0 LPA and 18:2 LPA) correlated with the age at onset of CUD in male patients; (iii) plasma concentrations of LPA species were affected by sex, showing that women had higher concentrations of all LPA species (except for 18:0 LPA) than men; (iv) plasma concentrations of LPA species changed in CUD patients with comorbid anxiety (lower total LPA, 18:0 LPA and 20:4 LPA), personality (higher 18:1 LPA and 20:4 LPA) and attention deficit hyperactivity (lower total LPA and 16:0 LPA) disorders; and (v) there were positive and significant correlations between total LPA, 16:0 LPA and 18:2 LPA species with KYNA as well as a positive correlation between 18:1 LPA and TRP. Overall, these data suggest that there is strong sexual dimorphism in plasma LPA concentrations, and its levels could play a role in anxiety responses in CUD patients with comorbid psychiatric disorders.

## Figures and Tables

**Figure 1 ijms-24-15586-f001:**
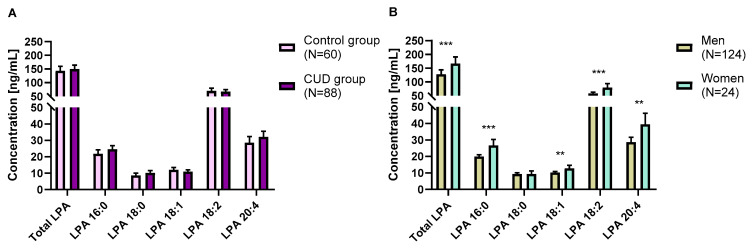
Plasma concentrations of total LPA and its LPA species in the sample according to the group (**A**) and sex (**B**). Bars are estimated marginal means and 95% confidence intervals (95%) representing LPA species (ng/mL) according to the group. Data were analyzed using two-way analysis of covariance (ANCOVA). ** *p* < 0.010 and *** *p* < 0.001 denote a significant main effect of group factor or sex. Abbreviations: CUD = cocaine use disorder, LPA = lysophosphatidic acid.

**Figure 2 ijms-24-15586-f002:**
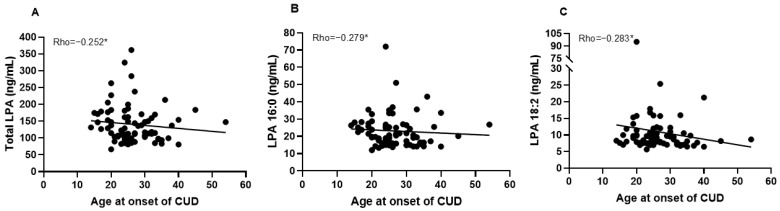
Correlations between total LPA (**A**), 16:0 LPA (**B**) and 18:2 LPA (**C**) with age at onset of CUD in male patients controlled by age and BMI. * *p* < 0.05 denotes a significant correlation. Abbreviations: CUD = cocaine use disorder, LPA = lysophosphatidic acid.

**Figure 3 ijms-24-15586-f003:**
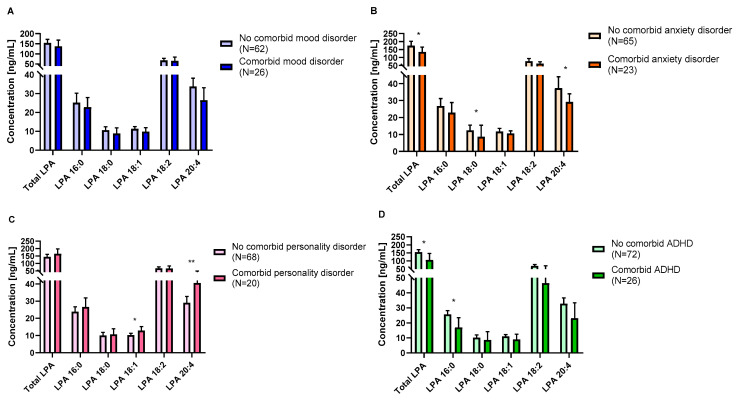
Plasma concentrations of LPA species according to psychiatric comorbidity for mood disorder (**A**), anxiety disorder (**B**), personality disorder (**C**) and ADHD (**D**). Bars are estimated marginal means and 95% confidence intervals (95%) representing LPA species (ng/mL) according to the group. Data were analyzed using two-way analysis of covariance (ANCOVA). * *p* < 0.05 and ** *p* < 0.010 denote a significant main effect of group factor. Abbreviations: ADHD = attention deficit hyperactivity disorder (childhood).

**Figure 4 ijms-24-15586-f004:**
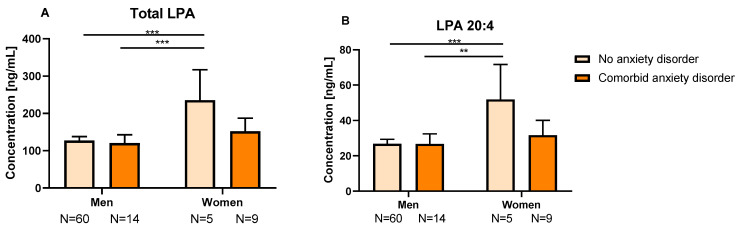
Plasma concentrations of total LPA (**A**) and 20:4 LPA (**B**) depending on comorbid anxiety disorder and sex. Bars are estimated marginal means and 95% confidence intervals (95%) representing LPA species (ng/mL) according to the group. Data were analyzed using two-way analysis of covariance (ANCOVA). ** *p* < 0.01 and *** *p* < 0.001 denote a significant interaction between group and sex factors. Abbreviation: LPA = lysophosphatidic acid.

**Figure 5 ijms-24-15586-f005:**
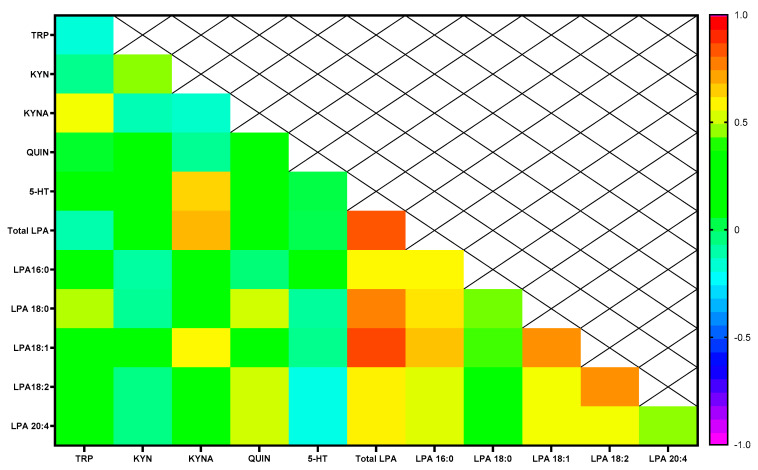
Correlation analysis between plasma concentrations of LPA species (ng/mL) and tryptophan metabolites [TRP (nmol/mL), KYN (nmol/mL), KYNA (pmol/mL), QUIN (pmol/Ml) and 5-HT (pmol/mL)]. Colors show partial correlation coefficient controlling for age and BMI. Abbreviations: LPA= lysophosphatidic acid, TRP = tryptophan, 5-HT = serotonin, KYN = kynurenine, QUIN = quinolinic acid, KYNA= kynurenic acid.

**Table 1 ijms-24-15586-t001:** Socio-demographic characteristics of the total sample.

Total Sample N = 148					
Variables		Control Group N = 60	CUD Group N = 88	Statistic	*p*-Value
Age ^1^ (Mean ± SD)	Years	37.52 ± 1.20	37 ± 8.43	2267.500	0.143
Body Mass Index ^1^ (Mean ± SD)	kg/m^2^	25.65 ± 3.51	25.98 ± 4.80	2582.500	0.991
Sex ^2^ [N (%)]	Women Men	10 (16.7) 50 (83.3)	14 (15.9) 74 (84.1)	0.015	0.902
Education Degree ^2^ [N (%)]	Elementary Secondary University	1 (1.70) 35 (58.30) 24 (40)	17 (19.30) 58 (65.90) 13 (14.80)	18.547	**<0.001**
Occupation ^2^ [N (%)]	Employed Unemployed Other	59 (98.30) - 1 (1.7)	35 (39.80) 7 (8) 46 (52.3)	49.500	**<0.001**

^1^ Value was calculated with Mann–Whitney U test. ^2^ Value was calculated with Fischer’s exact test or chi-squared test. Bold values are statistically significant for *p* < 0.05. Abbreviation: CUD = cocaine use disorder.

## Data Availability

Not applicable.

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
