# Peer review of "Plasma Lysophosphatidic Acid Concentrations in Sex Differences and Psychiatric Comorbidity in Patients with Cocaine Use Disorder"

_ijms, 2023, doi:10.3390/ijms242115586_

Round 1
Reviewer 1 Report
Dear Authors, I had the opportunity to review your study, which I read and largely appreciated, since the issues have been effectively and scientifically stated. The methodology is sound, as far as I have been able to determine, while highlighting the limitations which, to be fair, have been faithfully described by you. I suggest that the biological mechanisms of neurochemical alterations associated with cocaine use disorder should be spelled out even more precisely. This could have several implications for patients in order to provide useful information on the development of new treatments aimed at modulating levels of lysophosphatidic acid or its receptors in an attempt in order to influence symptoms and progression of the disorder. In addition, you are asked to specify whether your study showed differences in lysophosphatidic acid plasma concentrations based on psychiatric comorbidity, a circumstance that in itself could be useful in order to understand the interactions between cocaine use disorder and other mental conditions.
This could influence the overall treatment approach by considering both cocaine use disorder and psychiatric comorbidity in the treatment plan, identifying new therapeutic targets.
Addressing these elements of discussion in detail would certainly help make the article more comprehensive, which would be advisable, since the article has qualities and strengths that make it an acceptable scientific research contribution.
Author Response
Reviewer 1
Dear Authors, I had the opportunity to review your study, which I read and largely appreciated, since the issues have been effectively and scientifically stated. The methodology is sound, as far as I have been able to determine, while highlighting the limitations which, to be fair, have been faithfully described by you.
Response: we are very thankful for your suggestions and contributions to improve the quality of the article. We have tried to include all the improvement proposals and we think that the article is more completed now.
I suggest that the biological mechanisms of neurochemical alterations associated with cocaine use disorder should be spelled out even more precisely. This could have several implications for patients in order to provide useful information on the development of new treatments aimed at modulating levels of lysophosphatidic acid or its receptors in an attempt in order to influence symptoms and progression of the disorder.
Response: Thank you very much for your suggestion, it is true that there was a lack of information to understand the neurobiological bases of cocaine use disorder. That is why we have included various topics in the introduction:
- The relationship between the neurobiology of addiction (dopamine) and plasma concentrations of LPA: it is known that cocaine addiction produces an altered production of dopamine, and it is known that the activation of LPA receptors is associated with the release of catecholamines (106-109).
“On the other hand, because the activation of LPA receptor has been involved with the release of neurotransmitters (e.g., dopamine) associated with behavior, reward and learning and memory processes, it has been suggested to be implicated in substance use disorders (30–33)”
- Neurobiology of sexual differences between men and women who have cocaine use disorder: it is known that women are more vulnerable to the reinforcing effects of cocaine and that they are more predisposed to developing psychiatric disorders. However, these differences may be explained in part by gonadal hormones and by chromosomal mechanisms (X, Y) (68-73 lines).
“Furthermore, sex differences between men and women with cocaine abuse has been widely reported (12,13). Women seem to be more vulnerable to the reinforcing effects of cocaine as well as the development of depression, anxiety, coping with stress, and ac-tivation of the hypothalamic-pituitary-adrenal (HPA) axis. Effects that can be partially explained by biological differences in male and female gonadal hormones and in chromosomal mechanisms (X and Y) (14).”
The results of our study could suggest:
- LPA sexual dimorphism: LPA concentrations could be linked to female hormones or differences in the Y chromosome (257-262 lines).
“Therefore, Autotaxin-LPA pathway could develop an essential role in endocrine function since several studies have proposed them as potential biomarkers of hepatic steatosis in obesity (52) as well as in breast, ovarian and endometrial cancer in women (53–55). Therefore, LPA concentrations could be linked to female hormones or differ-ences in the Y chromosome that could explain their psychiatric vulnerability especially in anxiety disorders.”
- Potential pharmacology of LPA: since women with anxiety have similar levels to men (low), increasing LPA concentrations could be a promising treatment to alleviate symptoms of stress and anxiety (lines 401-402).
“Therefore, since women with anxiety disorders have lower levels of plasma LPA concentrations, pharmacological therapies aimed at increasing its levels could be a promising treatment to alleviate symptoms of stress and anxiety.”
- Explanation of why men consume more alcohol and other drugs of abuse: animal studies have indicated that LPA1-null mice with high anxiety use alcohol to mitigate their symptoms. Therefore, the low concentrations of LPA that we found in men could explain their vulnerability to developing substance use disorders (403-408 lines).
“In this regard, because of preclinical studies have demonstrated that LPA1 knockout mice consume alcohol for mitigate their anxiety (71), men and women with low circu-lant levels of LPA may be particularly vulnerable to alcohol, tobacco, cannabis and hypno-sedative addiction. In addition, dimorphism in plasma LPA concentrations (men have lower levels) could also explain why men have higher rates of alcohol and cannabis use disorders than women.”
In addition, you are asked to specify whether your study showed differences in lysophosphatidic acid plasma concentrations based on psychiatric comorbidity, a circumstance that in itself could be useful in order to understand the interactions between cocaine use disorder and other mental conditions. This could influence the overall treatment approach by considering both cocaine use disorder and psychiatric comorbidity in the treatment plan, identifying new therapeutic targets. Addressing these elements of discussion in detail would certainly help make the article more comprehensive, which would be advisable, since the article has qualities and strengths that make it an acceptable scientific research contribution.
Response: This is a very interesting question. However, we found no differences in total LPA concentrations or its species as a function of psychiatric comorbidity itself (see below). It is important to mention that our psychiatric comorbidity variable includes a wide variety of mental disorders (depressive, anxiety, psychotic, eating behavior, personality) that seem to be unrelated to LPA concentrations, so its significance disappears. However, this is a preliminary study and much more sample is needed to establish more robust associations. Therefore, we will take into account the variable of psychiatric comorbidity in LPA concentrations as a possible treatment target in future lines of research.
|
Pruebas de efectos inter-sujetos |
|||||
|
Variable dependiente: LogLPA_Total |
|||||
|
Origen |
Tipo III de suma de cuadrados |
gl |
Media cuadrática |
F |
Sig. |
|
Modelo corregido |
,303a |
5 |
,061 |
3,101 |
,013 |
|
Intersección |
9,463 |
1 |
9,463 |
483,678 |
,000 |
|
Age |
,001 |
1 |
,001 |
,063 |
,803 |
|
BMI |
,000 |
1 |
,000 |
,014 |
,906 |
|
Psychiatric_comorbidity |
,024 |
1 |
,024 |
1,246 |
,268 |
|
Sex |
,279 |
1 |
,279 |
14,281 |
,000 |
|
Psychiatric_comorbidity * Sex |
,029 |
1 |
,029 |
1,488 |
,226 |
|
Error |
1,604 |
82 |
,020 |
|
|
|
Total |
398,868 |
88 |
|
|
|
|
Total corregido |
1,908 |
87 |
|
|
|
|
a. R al cuadrado = ,159 (R al cuadrado ajustada = ,108) |
|||||

Reviewer 2 Report
Review-
1. The introduction briefly mentions the association between psychiatric comorbidity and CUD but does not clearly state the research gap or the specific questions the study aims to address. It would be helpful to explicitly state the objectives or hypotheses of the study.
2. The introduction mentions sex differences in CUD but could provide a bit more context or literature background on why studying these differences is important in the context of CUD.
3. The correlation analysis presented in the text appears somewhat limited. It's important to discuss the strength and significance of correlations. Additionally, providing context or hypotheses related to these correlations can help readers understand their relevance.
4. The discussion should provide an in-depth interpretation of the results presented in the preceding section. While some interpretations are provided, there are areas where the results are not discussed in detail. For example, the associations between LPA species and personality disorders are mentioned, but the implications of these associations are not explored.
Author Response
Reviewer 2
- The introduction briefly mentions the association between psychiatric comorbidity and CUD but does not clearly state the research gap or the specific questions the study aims to address. It would be helpful to explicitly state the objectives or hypotheses of the study.
Response: we are glad to this suggestion. We believe that remark the main objectives of the study at the end of the introduction could help readers better understand the purpose of the article. We have included that in lines 138-144.
“Therefore, the main objectives of the present observational study were to investigate: 1) sex differences in cocaine-related variables and psychiatric comorbidity, 2) the impact of cocaine abuse on plasma concentrations of LPA species compared to controls, 3) sex differences in plasma concentrations of LPA species (sexual dimorphism), 4) the differences in plasma concentrations of LPA species among psychiatric disorders in CUD patients, 5) the associations between plasma levels of LPA species and stress-related factors such as cortisol and TRP metabolites (TRP, KYN, QUIN, KYNA, 5-HT).”
- The introduction mentions sex differences in CUD but could provide a bit more context or literature background on why studying these differences is important in the context of CUD.
Response: we agree with your suggestion about talking more about gender differences in CUD and why this is relevant. That is why we have added more information in that paragraph on lines 68-72, 77-81, 116-127. In this way, we want to imply that women with CUD are more sensitive to the reinforcing effects of cocaine and are more vulnerable to the development of psychiatric disorders, due in part to neurobiological and social conditions. Therefore, given that different therapeutic strategies are necessary in men and women due to their different biological and social conditions, it is necessary to focus on sex differences in substance use disorders to create new effective treatments. That is why we have chosen the LPA biomolecule since there is strong sexual dimorphism in humans, it is related to anxious behaviors in animals and there is a high LPA1 receptor density in emotion-related regions (frontal cortex, hippocampus, amygdala and striatum) (all of this is explained in the introduction and in the discussion section). As the relationship between LPA and anxiety has not been demonstrated in humans, we wanted to investigate the correlations of LPA with other known anxiety markers such as cortisol and tryptophan metabolites. That is why we have also included gender differences in these biomarkers in the article. In addition, we have changed the order of the paragraphs in the introduction to better understanding.
“Furthermore, sex differences between men and women with cocaine abuse has been widely reported (12,13). Women seem to be more vulnerable to the reinforcing effects of cocaine as well as the development of depression, anxiety, coping with stress, and ac-tivation of the HPA axis. Effects that can be explained by biological differences in male and female gonadal hormones and in chromosomal mechanisms (X and Y) (14). Thus, despite women have lower rates of cocaine use and attend treatment centers less than men, they suffer more anxiety, mood and eating disorders (13,15). However, antisocial personality disorders and other concomitant substance use disorders -including to-bacco, alcohol and cannabis- are more common among men with CUD (6,16). In addition, women suffer more social discrimination and worse educational levels, unemployment rates, socioeconomic status, and health conditions. Instead, men have a higher prevalence of legal or criminal problems (13,17,18). Therefore, given that different therapeutic strategies are necessary in men and women due to their different biological and social conditions, it is necessary to focus on sex differences in substance use dis-orders to create new effective treatments (12,18,19).”
“In this context, stress is the main precursor of affective disorders vulnerability through the activation of the hypothalamic-pituitary-adrenal (HPA) axis and the release of glucocorticoids (35). The adaptative function of cortisol response consists of a rapid rise and decline following an environmental stressor. Prolonged or excessive cortisol response due to repetitive stress, especially in early stages of life, may lead to the de-velopment of a mental disorder (36). Paradoxically, low cortisol levels are shown in psychiatric conditions such as major depression, bipolarity, anxiety and schizophrenia (37–39). However, cortisol levels might change in opposite directions depending on the subtype of depressive disorder (melancholic vs atypical/psychotic) or in the phase of bipolarity (mania/irritability)(37). In addition, there are sex differences in cortisol changes: while woman with major depressive disorder or anxiety disorder have atten-uated cortisol stress response, men with major depressive disorder and social anxiety disorder exhibit an augmented cortisol response (39). On the other hand, increased corticosteroids concentrations have been associated with the dysregulation of the kynurenine pathway, by which the amino acid tryptophan (TRP) is metabolized (40–42).”
- The correlation analysis presented in the text appears somewhat limited. It's important to discuss the strength and significance of correlations. Additionally, providing context or hypotheses related to these correlations can help readers understand their relevance.
We understand this suggestion. That is why we have tried to deepen the analyzes by analyzing the correlations between the LPA species with the variables related to addiction in men and women separately (see new figure 2, 207-2013 lines). In this way we have realized that we only found significant correlations between the total concentration of LPA and some species in men but not in women. Even so, it is true that the correlations are still slight. The explanation we give to these correlations is that, as LPA is positively correlated with BDNF (a neuroprotective biomarker), there could be a compensatory response in those who develop problematic cocaine use earlier compared to those who develop the addiction later (342-347 lines).
“However, we found that 18:2 LPA was negatively correlated with the age of CUD diagnosis in the total sample, with total LPA, 16:0 LPA and 18:2 LPA being correlated with the age of CUD onset especially in men. This result may indicate a compensatory response against early development of problematic cocaine use since LPA species have been considered as a neuroprotective signaling that correlates with Brain-Derived Neurotrophic Factor (BDNF) in abstinent alcohol patients (28)”.
- The discussion should provide an in-depth interpretation of the results presented in the preceding section. While some interpretations are provided, there are areas where the results are not discussed in detail. For example, the associations between LPA species and personality disorders are mentioned, but the implications of these associations are not explored.
Response: We agree with the reviewer regarding the proposed interpretations. The reasons why we have not explained the direct implications of this study is because we wanted to be cautious with respect to these statements since the results still have many limitations and more studies are necessary to confirm these conjectures. However, the main statements of this paper that you can found in the discussion are:
- There is a robust LPA sexual dimorphism: LPA concentrations could be linked to female hormones or differences in the Y chromosome that could explain their psychiatric vulnerability especially in anxiety disorders (257-262 lines)
“Therefore, Autotaxin-LPA pathway could develop an essential role in endocrine function since several studies have proposed them as potential biomarkers of hepatic steatosis in obesity (52) as well as in breast, ovarian and endometrial cancer in women (53–55). Therefore, LPA concentrations could be linked to female hormones or differ-ences in the Y chromosome that could explain their psychiatric vulnerability especially in anxiety disorders.”
- Potential pharmacology of LPA: since women with anxiety have similar levels to men (low), increasing LPA concentrations could be a promising treatment to alleviate symptoms of stress and anxiety (lines 401-402) .
“Therefore, since women with anxiety disorders have lower levels of plasma LPA concentrations, pharmacological therapies aimed at increasing its levels could be a promising treatment to alleviate symptoms of stress and anxiety.”
- Explanation of why men consume more alcohol and other drugs of abuse: animal studies have indicated that LPA1-null mice with high anxiety use alcohol to mitigate their symptoms. Therefore, the low concentrations of LPA that we found in men could explain their vulnerability to developing substance use disorders (403-408 lines).
“In this regard, because of preclinical studies have demonstrated that LPA1 knockout mice consume alcohol for mitigate their anxiety (71), men and women with low circu-lant levels of LPA may be particularly vulnerable to alcohol, tobacco, cannabis and hypno-sedative addiction. In addition, dimorphism in plasma LPA concentrations (men have lower levels) could also explain why men have higher rates of alcohol and cannabis use disorders than women.”
Round 2
Reviewer 2 Report
Please accept the paper in present form.